# The Role of H_2_S Regulating NLRP3 Inflammasome in Diabetes

**DOI:** 10.3390/ijms23094818

**Published:** 2022-04-27

**Authors:** Huijie Zhao, Huiyang Liu, Yihan Yang, Honggang Wang

**Affiliations:** 1School of Basic Medical Sciences, Henan University, Kaifeng 475004, China; zhj5696@163.com (H.Z.); m15736875597@163.com (H.L.); 1323240458@163.com (Y.Y.); 2School of Nursing and Health, Henan University, Kaifeng 475004, China

**Keywords:** hydrogen sulfide, NLRP3 inflammasome, diabetic cardiomyopathy, diabetes-accelerated atherosclerosis, diabetic retinopathy

## Abstract

Nucleotide-binding oligomeric domain (NOD)-like receptor protein 3 (NLRP3) is a recently discovered cytoplasmic multiprotein complex involved in inflammation. The NLRP3 inflammasome contains NLRP3, apoptosis-related specific protein (ASC) and precursor caspase-1. The NLRP3 inflammasome is involved in many diseases, including diabetes. H_2_S is a harmful gas with a rotten egg smell. Recently, it has been identified as the third gas signal molecule after nitric oxide and carbon monoxide. It has many biological functions and plays an important role in many diseases, including diabetes. In recent years, it has been reported that H_2_S regulation of the NLRP3 inflammasome contributes to a variety of diseases. However, the mechanism has not been fully understood. In this review, we summarized the recent role and mechanism of H_2_S in regulating the NLRP3 inflammasome in diabetes, in order to provide a theoretical basis for future research.

## 1. Introduction

In 2002, Tschopp et al., first proposed a definition of inflammasome [1]. An inflammasome is a complex composed of a variety of proteins. It is an important part of the innate immune system, and can detect metabolic alarms, pathogens and infections in cells [2,3]. At present, there are five kinds of inflammasomes: NLRP1, NLRP3, NLRC4, AIM2 and RIG-I [4,5,6]. The NLRP3 inflammasome, consisting of NLRP3, apoptosis-associated spot-like protein (ASC) and a precursor of caspase-1, is the most deeply studied inflammasome [7,8,9,10,11,12]. NLRP3 has a molecular weight of 115 kDa, and mainly exists in monocytes, lymphocytes, epithelial cells, dendritic cells, osteoblasts and neutrophils. It contains three domains: one domain is the N-terminal pyran domain (PYD), which can recruit ASC; another domain is the central nucleotide-binding oligomeric domain NACHT, which possesses ATPase activity; and the third domain is the C-terminal leucine-rich repeat (LRR) [13,14]. ASC contains the carboxyl terminal CARD and the amino terminal PYD. When the NLRP3 inflammasome is activated, ASC interacts with NLRP3 through the PYD domain, on the one hand, and recruits pro-caspase-1 through a CARD–CARD domain interaction, on the other hand [5,15,16]. When the body is stimulated by exogenous or endogenous stimuli, NLRP3 is activated. The activated NLRP3 interacts with ASC and pre-caspase-1 to form a large protein complex, so as to activate pre-caspase-1. The activated caspase-1 converts pre-IL-1β and pre-IL-18 into their active form to induce inflammation [17]. The activation process of the NLRP3 inflammasome can be divided into two steps, which are mediated by the first and second signals, respectively. The first signal (signal 1) represents tissue injury or infection, including Toll-like receptor 4 (a pattern recognition receptor), which can recognize lipopolysaccharides (LPS), and a series of endogenous risk signals to activate NF-ĸB to increase the expression of the NLRP3 inflammasome, pro-IL-1β and pro-IL-18 [18,19]. Then, the second activation signal (signal 2) represents cell damage, including urate and cholesterol crystals and extracellular adenosine triphosphate (ATP), and induces the assembly of the inflammasome and the autolysis of pro-caspase-1. The activated caspase-1 converts the precursors IL-1β and IL-18 into their active form (Figure 1) [20]. NLRP3 inflammasome dysfunction can contribute to many diseases, including diabetes [21].

Hydrogen sulfide (H_2_S) has long been regarded as a toxic pollutant gas, but, recently, it has been considered as the third gas signal molecule after nitric oxide (NO) and carbon monoxide (CO) [22,23,24]. There are three enzymes that catalyze the synthesis of endogenous H_2_S: cystathionine-γ-lyase (CSE), cystathionine-β-synthase (CBS) and 3-mercaptopyruvate thiotransferase (3-MST) [25,26]. During the generation of endogenous H_2_S, L-cystathionine is produced by the β-substitution reaction of homocysteine with serine, catalyzed by CBS. The elimination of α- and γ-cysteine in L-cystathionine catalyzed by CSE produces L-cystenine. L-cystenine is then catalyzed by CSE/CBS to produce H_2_S via a β-elimination reaction. L-cystenine is also catalyzed by cysteine aminotransferase (CAT) to produce 3-mercaptopyruvate (3-MP), by transferring its amines to α-ketoglutarate. 3-MP is catalyzed by 3-MST to be converted into H_2_S (Figure 2) [27,28]. H_2_S contributes to many physiological processes, including anti-apoptosis [29], antioxidant stress [30], anti-inflammation [31], vasodilation, decreased blood pressure [32,33], cell survival/death, cell differentiation, cell proliferation/hypertrophy and mitochondrial bioenergy/biogenesis [34]. In recent years, more and more evidence has shown that exogenous H_2_S regulates the NLRP3 inflammasome and plays a vital role in a variety of pathological processes [24]. In this review, we summarized the recent research on H_2_S regulation of the NLRP3 inflammasome in diabetes to provide a theoretical reference for future related research.

## 2. The Role of H_2_S in Regulating the NLRP3 Inflammasome in Diabetic Cardiomyopathy

Diabetic cardiomyopathy (DC) is characterized by early diastolic dysfunction. After that, heart failure occurs without coronary artery disease, hypertension and dyslipidemia. The pathogenesis of diabetic cardiomyopathy is closely related to insulin resistance, hyperinsulinemia and hyperglycemia. The pathophysiological factors of diabetes include systemic metabolic disorder, oxidative stress, inflammation and dysfunctional immune regulation, which promote cardiac interstitial fibrosis, cardiac stiffness/diastolic dysfunction and subsequent systolic dysfunction, thus inducing clinical heart failure syndrome and causing cardiomyopathy [35,36,37]. Diabetic cardiomyopathy is one of the main causes of death in diabetic patients. At present, specific strategies for its prevention and treatment have not yet been elucidated [38,39]. Apoptosis, inflammation and oxidative stress play an important role in diabetic cardiomyopathy [40,41,42]. The results of the study conducted by Zena Huang et al., showed that high glucose (HG) promoted apoptosis by increasing the level of caspase-3 protein and induced inflammation by upregulating the level of IL-1β and IL-18 in H9c2 cells, which was abolished by exogenous H_2_S, suggesting that exogenous H_2_S could improve HG-induced cardiotoxicity. HG also induced mitochondrial membrane potential (MMP) loss and ROS expression, and downregulated the expression of the NLRP3 inflammasome, IL-1β and IL-18, which was abolished by exogenous H_2_S. The in-depth research showed that H_2_S suppressed the activation of TLR4 and NF-κB, induced by HG. NLRP3 gene silencing with siRNA significantly reduced HG-induced cell apoptosis, inflammation, the ROS level and MMP loss, indicating that the NLRP3 inflammasome mediated the H_2_S improvement in cardiotoxicity induced by HG. In HG-treated H9c2 cells, pretreatment with TAK-242 (TLR4 inhibitor) reduced the HG-induced phosphorylation of NF-κB; furthermore, treatment with BAY11-7082 (NF-κB inhibitor) downregulated the protein level of the NLRP3 inflammasome, prompting that HG promoted NLRP3 inflammasome activation through the TLR4/NF-κB pathway. Collectively, exogenous H_2_S improved HG-induced cardiotoxicity by suppressing NLRP3 inflammasome activation, by inhibiting the TLR4/NF-κB pathway in H9c2 cells [43]. It can be observed from the above study that H_2_S inhibits apoptosis by inhibiting the NLRP3 inflammasome in cardiomyocytes, which is involved in only a few studies. Further research is needed to clarify the relationship and mechanism between H_2_S, NLRP3 and apoptosis.

Low chronic inflammation in the myocardium is one of the main causes of diabetic cardiomyopathy [44,45]; hence, the inhibition of NLRP3 inflammasome-mediated inflammation can effectively improve diabetic cardiomyopathy [46,47]. Qiang Jia et al., used streptozotocin to establish a type 1 diabetic rat model, and found that H_2_S improved DC by ameliorating cardiac hypertrophy, enhancing the systolic and diastolic capacity of the heart, and alleviating histopathological and ultrastructural lesions in diabetic rats. Further research revealed that exogenous H_2_S also notably reduced the serum level of cardiac enzymes and pro-inflammatory factors in the DC rat model, indicating that H_2_S could dampen cardiac damage and inflammation in DC. Hyperglycemia induced redox perturbation by upregulating TXNIP protein expression and downregulating thioredoxin protein expression in DC, which was mitigated by H_2_S. H_2_S inhibited NLRP3 inflammasome activation in DC, whereas PAG (an inhibitor of CSE) counteracted the above protective effects of exogenous H_2_S. Overall, exogenous H_2_S alleviated myocardial inflammation by suppressing NLRP3 inflammasome activation in diabetic rats, which needs to be further demonstrated using NLRP3 inhibitors [48]. Thioredoxin is an important antioxidant in the body [49,50]. TXNIP has been reported to promote oxidative stress by inhibiting thioredoxin [51,52]. Moreover, TXNIP is an important regulator of the NLRP3 inflammasome [53,54]. Therefore, it can be deduced that H_2_S inhibits NLRP3 inflammasome activation in diabetic rats through the suppression of TXNIP, although this requires further verification [48].

There is increasing evidence that necroptosis contributes to DC [46,55,56]. In order to further study the mechanism of necroptosis in the pathogenesis of diabetic cardiomyopathy, Weiwei Gong et al., carried out a series of experiments and found that myocardial H_2_S production, CSE mRNA expression and plasma H_2_S levels were significantly downregulated in STZ-induced diabetic mice. CSE deficiency aggravated DC and promoted oxidative stress, necroptosis and the NLRP3 inflammasome. Similarly, the CSE inhibitor (PAG) promoted cell injury and oxidative stress, and promoted cardiomyocyte necroticptosis and NLRP3 inflammasome activation in cardiomyocytes induced by HG. However, in vivo and in vitro, exogenous H_2_S significantly improved diabetic cardiomyopathy, alleviating oxidative stress, necroptosis and the NLRP3 inflammasome. In conclusion, exogenous H_2_S could improve DC by inhibiting necroticptosis, the NLRP3 inflammasome and oxidative stress [57]. It has been reported that excessive oxidative stress is correlated with necroptosis [58,59,60]. Necroptosis also mediates the activation of the NLRP3 inflammasome [61]. Therefore, in the above study, it can be deduced that exogenous H_2_S inhibits the NLRP3 inflammasome by suppressing necroptosis [57], although this requires further study.

## 3. The Role of H_2_S in Regulating the NLRP3 Inflammasome in Diabetes-Accelerated Atherosclerosis

Diabetes-accelerated atherosclerosis is one of the most common cardiovascular complications of diabetes. Compared with non-diabetic patients, it has a high incidence rate, early onset time, fast course and high mortality [62,63,64]. When examining atherosclerosis in a diabetic mouse model, H_2_S inhibited the formation of aortic root plaques, and downregulated the levels of vascular cell adhesion molecule 1 (VCAM1) and intercellular adhesion molecule 1 (ICAM1). Meanwhile, the inflammatory factors IL-1β, IL-6, TNF-α and MCP1 were notably downregulated by H_2_S. The mechanism research revealed that exogenous H_2_S inhibited NLRP3 inflammasome activation in diabetes-accelerated atherosclerosis conditions. Silencing NLRP3 with siRNA dampened the increase in ICAM1 and VCAM1, induced by HG and oxLDL, suggesting that H_2_S protected the endothelium by suppressing NLRP3 inflammasome activation. In summary, exogenous H_2_S significantly ameliorated diabetes-accelerated atherosclerosis by inhibiting NLRP3 inflammasome activation and oxidative stress, thus prompting that H_2_S inhibited NLRP3 inflammasome activation through the inhibition of oxidative stress in diabetes-accelerated atherosclerosis [65].

## 4. The Role of H_2_S in Regulating the NLRP3 Inflammasome in Gestational Diabetes Mellitus

Gestational diabetes mellitus (GDM) is the most common complication during pregnancy. It is characterized by insulin resistance and glucose intolerance, and is associated with adverse consequences for parturient women and newborns [66,67,68]. It has been reported that excessive activation of the NLRP3 inflammasome contributes to GDM [69,70,71]. The results of the study conducted by Wei Wu et al., revealed that the protein expression levels of NLRP3 and cleaved caspase-1 were upregulated, whereas the protein expression levels of CBS and CSE were downregulated in GDM placenta. Moreover, the CBS and CSE levels were negatively correlated with the levels of NLRP3 and cleaved caspase-1. The in-depth research showed that exogenous H_2_S and H_2_S precursor L-cysteine notably reduced the protein expression of NLRP3 and cleaved caspase-1, and reduced the levels of IL-1β and IL-18 in human placental cells, while the NLRP3 inflammasome inhibitor Ac-YVAD-CMK reduced the levels of IL-1β and IL-18. Overall, exogenous H_2_S inhibited NLRP3 inflammasome activation in GDM [72]. In the above studies, the mechanism of exogenous H_2_S regulating the NLRP3 inflammasome needs to be clarified.

## 5. The Role of H_2_S in Regulating the NLRP3 Inflammasome in Diabetic Adipose Tissue Inflammation

Adipose tissue is an important endocrine organ, which is involved in regulating inflammation [73,74,75]. The NLRP3 inflammasome plays an important role in inflammation in adipose tissue [76,77]. H_2_S has an anti-inflammatory function in many tissues and cells [78,79,80]; however, the anti-inflammatory properties of H_2_S in adipose tissue have not been reported. Tian-Xiao Hu et al., found that HG upregulated the expression of NLRP3, ASC, cleaved caspase-1, and the levels of IL-1β and IL-18 in adipocytes. The inhibition of caspase-1 with inhibitors could eliminate the increase in IL-1β and IL-18, induced by HG, in adipocytes. Exogenous H_2_S inhibited the induction of NLRP3, ASC, cleaved caspase-1, IL-1β and IL-18, induced by HG, in adipocytes. Overall, exogenous H_2_S inhibited NLRP3 inflammasome-mediated inflammation, induced by HG, in adipocytes [81]. At present, there are few studies on the role of H_2_S in regulating the NLRP3 inflammasome in adipose tissue, which is worthy of in-depth study; in particular, the relevant mechanism needs to be clarified.

## 6. The Role of H_2_S in Regulating the NLRP3 Inflammasome in Human Diabetic Retinopathy

Diabetic retinopathy (DR), one of the most common complications of diabetes, is the main cause of blindness [82,83,84]. It has been reported that oxidative stress and inflammation are closely related with DR [85,86,87,88]. Peng Wang et al., found that in ARPE-19 (human retinal pigment epithelial) cells, HG promoted cell death and upregulated the expression levels of IL-18 and IL-1β. Furthermore, HG also upregulated NLRP3 inflammasome expression and promoted ROS production, whereas the ROS scavenger NAC inhibited HG-induced ROS production, downregulated the expression levels of IL-18 and IL-1β, and abolished NLRP3 inflammasome activation, induced by FG. Moreover, silencing NLRP3 gene expression with siRNA had similar results to those with NAC. The above indicated that HG induced NLRP3 inflammasome-mediated inflammation by promoting ROS production. The in-depth study showed that exogenous H_2_S was able to completely abolish the increase in ROS production and NLRP3 inflammasome-mediated inflammation induced by HG in ARPE-19 cells. Collectively, exogenous H_2_S protected human retinal pigment epithelial cells against NLRP3 inflammasome-mediated inflammation, induced by HG, by inhibiting ROS production [89]. Contrary to the above conclusion, H_2_S can promote inflammation and elevate oxidative stress [90,91,92], which may be related to the concentration of H_2_S. A low concentration of H_2_S inhibits inflammation and oxidative stress and has a protective effect on the body, while a high concentration of H_2_S has the opposite effect.

## 7. The Role of H_2_S in Regulating the NLRP3 Inflammasome in Diabetic Fibrosis of the Diaphragm

The contractile function of the diaphragm in diabetic rats, induced by streptozotocin (STZ), decreased significantly, which was related to the excessive inflammation and oxidative stress in the diaphragm tissue [93,94]. The available evidence suggests that hyperglycemia can promote the deposition of collagen in muscle, which leads to fibrosis [95], and the excessive activation of the NLRP3 inflammasome contributes to tissue fibrosis [96,97,98]. Rui Yang et al., found that the biomechanical parameters of the diaphragm in diabetic rats were reduced, while the levels of inflammatory cytokines, NLRP3 inflammasome and collagen were upregulated, suggesting that diabetes induced diaphragm muscle fibrosis by activating the NLRP3 inflammasome. Exogenous H_2_S improved the histological and ultrastructural lesions, diaphragmatic biomechanical alterations and fibrosis of the diaphragm, and inhibited NLRP3 inflammasome activation in streptozotocin-induced diabetic rats. Collectively, exogenous H_2_S partly ameliorated the diaphragm muscle fibrosis of streptozotocin-induced diabetic rats by suppressing NLRP3 inflammasome activation, which needed to be further confirmed [99]. TGF-β1 is a contributor to tissue fibrosis [29]. Overactivation of the NLRP3 inflammasome can lead to the excessive production of TGF-β1, which can lead to fibrosis [100]. In the above study, how H_2_S regulates the NLRP3 inflammasome remains to be clarified.

## 8. Conclusions

Much evidence indicates that H_2_S plays an important role in diabetes. In this review, we summarized the role of H_2_S regulation in the NLRP3 inflammasome and its mechanism in many type of diabetes-related diseases, reaching the following conclusions: (1) exogenous H_2_S improves HG-induced cardiotoxicity through the inhibition of NLRP3 inflammasome activation, by suppressing the TLR4/NF-κB pathway in H9c2 cells; (2) H_2_S suppresses NLRP3 inflammasome activation in diabetic rats through the inhibition of TXNIP, which needs to be further verified; (3) exogenous H_2_S could improve diabetic cardiomyopathy through inhibition of the NLRP3 inflammasome, by inhibiting necroticptosis; (4) H_2_S mitigates diabetes-accelerated atherosclerosis through the inhibition of NLRP3 inflammasome activation, by suppressing oxidative stress; (5) exogenous H_2_S suppresses NLRP3 inflammasome activation in gestational diabetes mellitus; (6) exogenous H_2_S inhibits NLRP3 inflammasome activation induced by HG in adipocytes; (7) exogenous H_2_S inhibits the NLRP3 inflammasome, induced by HG, through the inhibition of ROS production in human retinal pigment epithelial cells; (8) exogenous H_2_S partly improves diaphragm muscle fibrosis through the inhibition of NLRP3 inflammasome activation in streptozotocin-induced diabetic rats, which needed to be further confirmed (Table 1).

It can be observed from the above that H_2_S may inhibit NLRP3 inflammasome activation through suppression of the TLR4/NF-κB pathway, TXNIP, necroticptosis and oxidative stress in diabetes, some of which need to be further confirmed. Whether there are other mechanisms of H_2_S inhibiting the NLRP3 inflammasome in diabetes requires further study. In addition, in the above studies, H_2_S played a role in diabetes by inhibiting the NLRP3 inflammasome. It has been reported that H_2_S has both anti-inflammatory and pro-inflammatory effects, so whether H_2_S can promote the NLRP3 inflammasome in diabetes also remains to be studied. The H_2_S/NLRP3 inflammasome has been involved in diabetes in many studies, respectively, but the roles and the mechanisms of the two in the occurrence and development of diabetes need to be further elucidated. Furthermore, the NLRP3 inflammasome is closely related to endoplasmic reticulum stress (ERS)/autophagy, and the latter two are the targets of H_2_S regulation. Therefore, the role of H_2_S in regulating the NLRP3 inflammasome/ERS or NLRP3 inflammasome/autophagy in diabetes is worth studying. Finally, the current donor of exogenous H_2_S is not ideal; therefore, the development of better H_2_S donors will be more conducive to the application of H_2_S in the regulation of the NLRP3 inflammasome in diabetes.

In conclusion, with the further development of related research, H_2_S regulation of the NLRP3 inflammasome will become a new strategy for the treatment of diabetes.

## Figures and Tables

**Figure 1 ijms-23-04818-f001:**
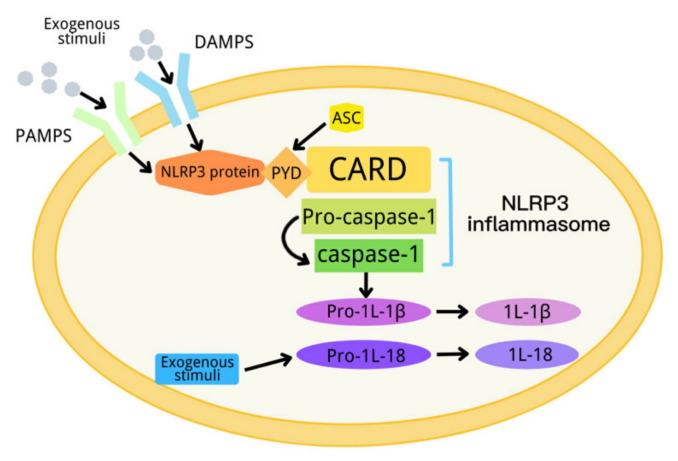
Schematic diagram of the NLRP3 inflammasome activation process.

**Figure 2 ijms-23-04818-f002:**
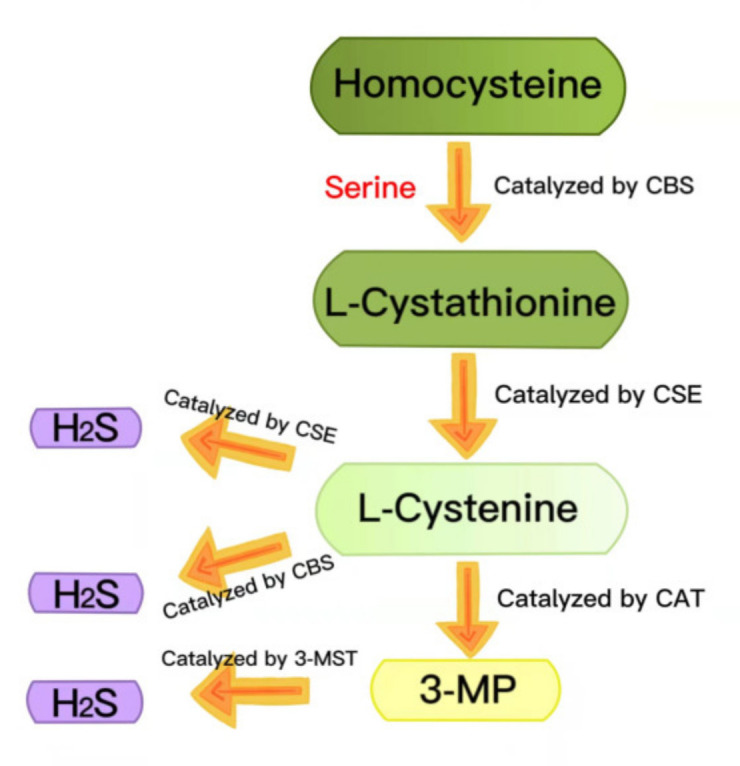
Summary of the production of endogenous H2S. CBS: cystathionine-beta-synthase; CSE: cystathionine-gamma-lyase; 3-MST: 3-mercaptopyruvate thiotransferase; 3-MP: 3-mercaptopyruvate; CAT: cysteine aminotransferase.

**Table 1 ijms-23-04818-t001:** Summary of the role of H_2_S in regulating the NLRP3 inflammasome in diabetes.

Type of Diabetes-Related Diseases	Role of Hydrogen Sulfide Regulation of NLRP3 Inflammasome	Experimental Materials	Reference
diabetic cardiomyopathy	exogenous H_2_S improves HG-induced cardiotoxicity through the inhibition of NLRP3 inflammasome activation by suppressing the TLR4/NF-κB pathway	high glucose (HG)-induced H9c2 cells	[43]
diabetic cardiomyopathy	H_2_S suppresses NLRP3 inflammasome activation in diabetic cardiomyopathy through the inhibition of TXNIP, which requires further verification	diabetic rats	[48]
diabetic cardiomyopathy	exogenous H_2_S improves diabetic cardiomyopathy through suppression of the NLRP3 inflammasome by inhibiting necroticptosis	diabetic mice/HG-induced cardiomyocyte	[57]
diabetes-accelerated atherosclerosis	H_2_S ameliorates diabetes-accelerated atherosclerosis through the inhibition of NLRP3 inflammasome activation by suppressing oxidative stress	HG-induced human umbilical vein endothelial cells/diabetic mice	[65]
gestational diabetes mellitus	exogenous H_2_S inhibits NLRP3 inflammasome activation in gestational diabetes mellitus	HG-induced human diabetic placenta tissues and placental cell	[72]
diabetic adipose tissue inflammation	exogenous H_2_S inhibits HG-induced NLRP3 inflammasome activation	HG-induced rat adipocytes	[81]
diabetic retinopathy	exogenous H_2_S inhibits the NLRP3 inflammasome induced by HG through the inhibition of ROS production	HG-induced human retinal pigment epithelium cell line ARPE-19	[89]
diabetic fibrosis of the diaphragm	exogenous H_2_S improves diaphragm muscle fibrosis partly through the inhibition of NLRP3 inflammasome activation	diabetic rats	[99]

## Data Availability

Not applicable.

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
