# Peer review of "The Role of H2S Regulating NLRP3 Inflammasome in Diabetes"

_ijms, 2022, doi:10.3390/ijms23094818_

Round 1

Reviewer 1 Report

Authors has summarized the recent role and mechanism of H2S in regulating NLRP3 inflammasome in diabetes, in order to provide a theoretical basis for the future researches. This paper well written and in my opinion paper should be accepted after minor revision.

Comments to the Author:

  1. The introduction should detail the aim and goals of the study by referring to the available literature studies and explaining how the undertaken studies would make these goals possible. It is not clearly expressed how to show H2S regulating NLRP3 inflammasome in diabetes diseases activity.
  2. Author should also explore which inflammasomes: NLRP1, NLRP3, NLRC4, AIM2 and RIG-I (4-6) is a most impotent for activity.
  3. Figure 1 and 2 is not clear, it should be revise in high resolution
  4. It is noted that the manuscript needs careful editing in English spelling and grammar.
  5. Conclusion: This part is also written poorly. The conclusion should include the most important findings and the outcomes thereof. The future perspectives are also expected to be mentioned in this part.

Author Response

1 The introduction should detail the aim and goals of the study by referring to the available literature studies and explaining how the undertaken studies would make these goals possible. It is not clearly expressed how to show H2S regulating NLRP3 inflammasome in diabetes diseases activity.

I'm sorry, we don't fully understand what you mean. We can only answer according to our understanding. In the introduction, we introduced NLRP3 inflammasome and H2S respectively, and pointed out that two of them played a role in diabetes. The purpose of this review is " In this review, we summerized the recent researches about H2S regulating NLRP3 inflammasome in diabetes to provide theoretical reference for the future related researches."

As to “how the undertaken studies would make these goals possible, how to demonstrate the role of H2S in regulating NLRP3 inflammasome in diabetes mellitus”, we have introduced in detail in part 2-7 of the manuscript. Those cannot be described in detail in the

introduction. We have made some modifications to the corresponding content in part 2-7 according to your opinion, so as to clarify the role of H2S in regulating NLRP3 inflammation in the activity of diabetes mellitus. If our answer doesn't satisfy you, please tell us and we'll change it again. Thank you!

2 Author should also explore which inflammasomes: NLRP1, NLRP3, NLRC4, AIM2 and RIG-I (4-6) is a most impotent for activity.

I'm sorry, we don't fully understand what you mean. At present, NLRP3 is the most thoroughly studied inflammasome, while there are few studies on other inflammasome. It is unclear which inflammasome has the worst function. If our answer is not satisfactory to you, please tell us and we will change it. Thanks you!

3 Figure 1 and 2 is not clear, it should be revise in high resolution

Yes! We’ve redrawn Figure 1 and Figure 2 in order to increase the resolution, thank you!

4 It is noted that the manuscript needs careful editing in English spelling and grammar.

Yes! We’ve carefully read the manuscript and revised the English spelling and grammar. See the text for details, thank you!

5 Conclusion: This part is also written poorly. The conclusion should include the most important findings and the outcomes thereof. The future perspectives are also expected to be mentioned in this part.

In conclusion, we have concluded the findings and results of the role of H2S regulating NLRP3 in diabetes. I have added "H2S/NLRP3 inflammasome has been involved in diabetes perspective in many studies, but the roles and the mechanisms of the two in the occurrence and development of diabetes needs to be further enhanced.". Together with previous statements, they are the future perspectives about the future research of H2S regulating NLRP3 in diabetes. If our answer doesn't satisfy you, please tell us and we'll change it again. Thank you!

Reviewer 2 Report

The paper entitled “The role of H2S regulating NLRP3 inflammasome in diabetes” comprises potentially significant data for the strategy of diagnostic, prognostic as well as therapeutic schedules particularly beneficial during for health service directed to management diabetic patients.

The Authors of this manuscript firmly emphasize that H2S regulation of NLRP3 inflammasome contributes to a variety of heterogenic disorders.

Remarks:

  1. The resolution of Figures no. 1, 2 is insufficient – Figures no. 1, 2 need to be redrawn;
  2. In the manuscript The Authors – probably not intentionally – omit the role of “H2S influence on NLRP3 inflammasome during pathogenesis and development of the diabetic retinopathy”.
  3. Moreover, it is absolutely necessary to answer the following question: can the H2S regulation NLRP3 inflammasome importantly influence the pathogenesis of diabetic foot ulcer development?
  1. Furthermore, the Authors should at least try to discuss the potential (if any) role of the “H2S influence on NLRP3 inflammasome during pathogenesis and development of the diabetic neuropathy”. Synthetic explanation is highly needed;
  2. The Authors should also try to discuss the H2S impact on NLRP3 inflammasome in cancer accompanied by the metabolic disease (with the special emphasize on carbohydrate metabolism disorders);
  3. The Conclusions lacks of any connections to the molecular aspects of diabetes molecular diagnostics.

Author Response

1 The resolution of Figures no. 1, 2 is insufficient – Figures no. 1, 2 need to be redrawn;

Yes! We’ve redrawn Figure 1 and Figure 2 in order to increase the resolution, thank you!

2 In the manuscript The Authors – probably not intentionally – omit the role of “H2S influence on NLRP3 inflammasome during pathogenesis and development of the diabetic retinopathy”.

We have described “the role of “H2S influence on NLRP3 inflammasome during

pathogenesis and development of the diabetic retinopathy” in part 6 of the review in part 6 “The role of H2S regulating NLRP3 inflammasome in human diabetic retinopathy”. If my answer doesn't satisfy you, please tell us and we'll change it. Thank you!

3 Moreover, it is absolutely necessary to answer the following question: can the H2S regulation NLRP3 inflammasome importantly influence the pathogenesis of diabetic foot ulcer development?

Thank you for your comment! There is no literature on H2S regulating NLRP3 on diabetes foot ulcers. This is also a topic worthy of future research. Thank you again!

4 Furthermore, the Authors should at least try to discuss the potential (if any) role of the “H2S influence on NLRP3 inflammasome during pathogenesis and development of the diabetic neuropathy”. Synthetic explanation is highly needed;

Thank you for your good advice. But I searched the internet carefully and found no literature on H2S regulating NLRP3 in diabetes neuropathy. Thank you again!

5 The Authors should also try to discuss the H2S impact on NLRP3 inflammasome in cancer accompanied by the metabolic disease (with the special emphasize on carbohydrate metabolism disorders);

Thank you for your good advice. This review is about the role of H2S in regulating NLRP3 in diabetes. I searched the internet carefully and found no literature on H2S regulating NLRP3 in in cancer accompanied by diabetes neuropathy. If my answer doesn't satisfy you, please tell us and we'll change it. Thank you again!

6 The Conclusions lacks of any connections to the molecular aspects of diabetes molecular diagnostics.

Thank you for your advice! This article summarizes the role and molecular mechanism of H2S regulating NLRP3 in diabetes mellitus, and has no relation with the diabetes molecular aspects of diabetes molecular diagnostics. Therefore, in the conclusion, we did not mention molecular diagnosis of diabetes. If my answer doesn't satisfy you, please tell us and we'll change it. Thank you again!

Round 2

Reviewer 2 Report

The Authors of the publication made appropriate corrections. They properly modified the manuscript. The article can be published without any corrections.